# Attribution Patching Outperforms
# Automated Circuit Discovery

## Abstract

Automated interpretability research has recently attracted attention as a potential research direction that could scale explanations of neural network behavior to large models. Existing automated circuit discovery work applies activation patching to identify subnetworks responsible for solving specific tasks (circuits). In this work, we show that a simple method based on attribution patching outperforms all existing methods while requiring just two forward passes and a backward pass. We apply a linear approximation to activation patching to estimate the importance of each edge in the computational subgraph. Using this approximation, we prune the least important edges of the network. We survey the performance and limitations of this method, finding that averaged over all tasks our method has greater AUC from circuit recovery than other methods.

## 1 Introduction

Mechanistic interpretability is a subfield of AI interpretability that focuses on attributing model behaviors to its components, thus reverse engineering the network (Olah, 2022). This field aims to identify subnetworks (circuits) within the model which are responsible for solving specific tasks (Olah et al., 2020). Prior attempts at finding circuits in language models have led to finding networks of attention heads and multi-layer perceptrons (MLPs) that partially or fully explain model behaviors at tasks such as indirect object identification, modular arithmetic, completion of docstrings, and predicting successive dates (Wang et al., 2023; Nanda et al., 2023; Heimersheim and Janiak, 2023; Hanna et al., 2023). However, almost all previous work has been limited to relatively small models since manually applying mechanistic interpretability methods has not currently scaled to end-to-end circuits in larger models (Lieberum et al., 2023).

It may be important to scale interpretability to large models as these are the neural networks most widely deployed and used by a wide range of people. Currently, we have little understanding into how these models work and failure modes are not always found ahead of deployment. If successful, scaled interpretability could address a wide variety of concerns about the lack of transparency of language models (Vig et al., 2020), in addition to speculative risks about the alignment of machine learning systems (Hubinger, 2020).

Automated Circuit Discovery (ACDC; Conmy et al. (2023)) attempts to automate a large portion of the mechanistic interpretability workflow — the pruning of edges between attention heads and MLPs that do not affect the task being studied. ACDC begins with a computational graph, and recursively calculates the importance of an edge in the graph for a specific task. In our work, we use edges to refer to activations inside models between two components (Section 2 describes this motivation further). ACDC's pruning algorithm applies **activation patching**. (Note that **activation patching** is not **attribution patching**. Both are defined in full in Section 3.3.) At a high level, activation patching edits a specific activation in a model forward pass and measures a model statistic (e.g loss) under this intervention. Activation patching is inefficient for circuit discovery because getting each statistic about model activations requires another forward pass. Our work uses **attribution patching** to recover circuits more efficiently (Section 3.3).

Our main contributions are:

1. Introducing a method for using attribution patching on all computational graph edges for automated circuit discovery (Edge Attribution Patching, Section 3.3).
2. Benchmarking Edge Attribution Patching vs existing circuit discovery methods (Section 4).
3. Finding and explaining some limitations with Edge Attribution Patching (Section 5).

## 2 Related Work

**Automated Circuit Discovery** refers to finding the important subgraph of models' computational graphs for performance on particular tasks (Conmy et al., 2023). Existing algorithms include efficient heuristics (Michel et al., 2019) and gradient-descent based methods (Louizos et al., 2018; Cao et al., 2021). ACDC is related to pruning (Blalock et al., 2020) and other compression techniques (Zhu et al., 2023), but differs in how the compressed networks are reflective of the circuits that model uses to compute outputs to certain tasks and the goal of ACDC is not to speed up forward passes (all techniques studied in this work slow forward passes).

**Activation Patching** is a technique for analyzing the role of individual components in a model. It involves targeted manipulations of activations during a forward pass (further explained in Section 3.1). Previous works applied this technique under various names, such as Interchange Interventions (Geiger et al., 2021), Causal Mediation Analysis (Vig et al., 2020) and Causal Tracing (Meng et al., 2022). We adapt the terminology used by Conmy et al. (2023).

**Transformer Circuits**. Our work builds upon the framework for understanding transformers for interpretability as introduced by Elhage et al. (2021). The important details include how they formulate forward passes of transformer models. Individual attention heads and MLPs (collectively called nodes) read and write information to a central communication channel, also called the residual stream. In these terms we can examine dependencies of nodes with the output of earlier nodes, i.e we can measure the effect of attention heads in layer 0 on the attention heads in layer 2. In the following, we view these dependencies as edges between nodes, building on existing work using this perspective (Heimersheim and Janiak, 2023; Hanna et al., 2023; Wang et al., 2023).

## 3 Edge Attribution Patching

We present **Edge Attribution Patching** (EAP) as a technique to identify relevant model components for solving a specific task. In the following, we view language models as directed, acyclic graphs. In these terms, we aim to find small subgraphs that retain good performance on narrow tasks. We determine the importance of a specific edge through targeted manipulation of activations during a forward pass. We compare two approaches, Attribu-

tion Patching and Activation Patching, in order to motivate EAP.

### 3.1 Activation Patching

*Activation patching* refers to replacing the activations from one model forward pass with the activations from a different forward pass. This method is typically applied to measure the counterfactual importance of model components, i.e. to measure a statistic $L(x)$ from model outputs under the activation patching, where $x$ is an input prompt. For example, $L$ often represents loss or logit difference (Wang et al., 2023).

Following existing work (Section 2), we study the effect of activation patching on specific model edges by setting these equal to activations from different forward passes. Concretely, suppose that an edge $E$ in the computational graph has activation $e_{\text{corr}}$ on some corrupted prompt. In this work, we use the change in metric under activation patching

$$|L(x_{\text{clean}}|\, \text{do}(E = e_{\text{corr}})) - L(x_{\text{clean}})| \quad (1)$$

to measure the impact of edge $E$. We use do-notation from causality (Pearl, 1995) to emphasise that activation patching is a causal intervention.

### 3.2 Attribution Patching

Activation patching slows ACDC since each measurement (like Equation (1)) requires another forward pass. *Attribution patching* (Nanda, 2023) is a technique for estimating Equation (1) for many different edges $E$ using only two forward passes and one backward pass.[1] It linearly approximates the metric difference after corrupting a single edge in the computational graph (Figure 1) by expanding $L$ as a function of the edge activation as a Taylor series with terms up to the first order:[2]

$$L(x_{\text{clean}}|\, \text{do}(E = e_{\text{corr}})) \approx L(x_{\text{clean}}) +$$
$$\underbrace{(e_{\text{corr}} - e_{\text{clean}})^\top \frac{\partial}{\partial e_{\text{clean}}} L(x_{\text{clean}}|\, \text{do}(E = e_{\text{clean}}))}_{\text{Call this } \Delta_e L, \text{ the } \textbf{attribution score}.}$$
$$(2)$$

---

[1] Attribution patching (like activation patching) also applies to nodes and other model internal components that aren't edges, but we only use edges in this work.

[2] Note that $L(x_{\text{clean}}|\, \text{do}(E = e_{\text{clean}})) = L(x_{\text{clean}})$ as all activations in this equation are from clean forward passes. We highlight the $e_{\text{clean}}$ since we take the gradient with respect to this activation.

A simple rearrangement implies that Equation (1) is approximately equal to $|\Delta_e L|$ (3) which we call the **absolute attribution score** for the rest of this paper. In this work we always compute this score across a set of $(x_{\text{clean}}, x_{\text{corr}})$ pairs and take the mean.

In practice, all gradients needed to calculate the attribution scores come from intermediate terms computed in one ordinary backwards pass[3] in Py-Torch

### 3.3 Edge Attribution Patching

We can use the insights from Section 3.2 to build an automated circuit discovery algorithm. This takes two steps: i) use Equation (2) to obtain absolute attribution scores for the importance of all edges in the computational graph, and then ii) sort these scores and keep the top $k$ edges in a circuit. We use **Edge Attribution Patching** (EAP) to refer to this algorithm. In the rest of the work we report results for all $k$ values when we evaluate EAP (similar to HISP in Conmy et al. (2023)).

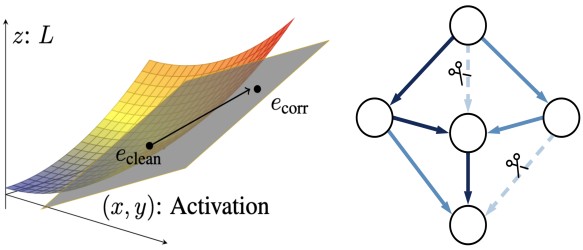

Figure 1: Edge Attribution Patching (EAP): (left) Attribution Patching approximates the difference in metric $L$ caused by corrupting edges. (right) Removing the least important edges.

Note that one limitation of attribution patching is that it will not work when the gradient of the metric is the zero vector. Conmy et al. (2023) recommended the use of KL divergence as a metric, which is i) equal to 0 when we run the model without ablations and ii) a non-negative metric. Therefore the zero point is a global minimum and hence all gradients are the zero vector at this point. In this work we use the task-specific metrics' (not KL divergence) from Conmy et al. (2023) so avoid this issue.

---

[3]In Appendix F we show how only one backwards pass is required.

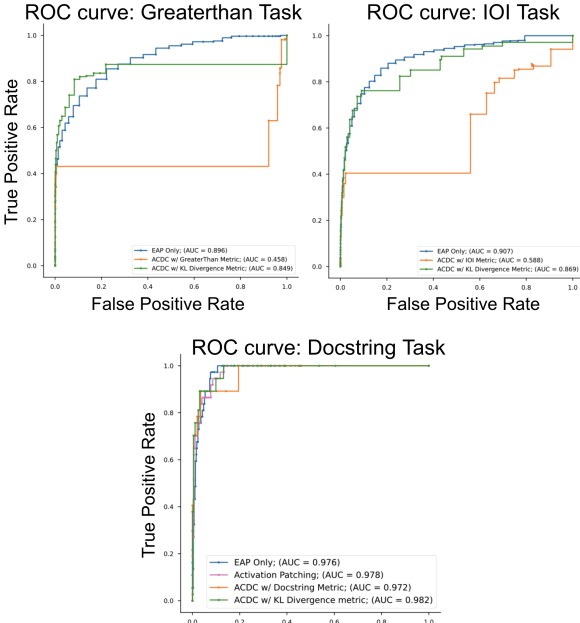

Figure 2: ROC Curves comparing EAP, ACDC with task metric, and ACDC with KL Divergence for the Greater-Than (left), IOI (right), and Docstring task (bottom). The Docstring plot also compares to Activation Patching.

## 4 Results

### 4.1 Edge Attribution Patching vs Activation Patching vs ACDC

We compare Edge Attribution Patching (EAP) and ACDC on the Indirect Object Identification (IOI), Docstring, and Greater-Than tasks. For each of these tasks, previous studies identified a subgraph (circuit) relevant for solving the task. We use their results as a ground truth for benchmarking both methods. We also compare using ACDC with the task-specific metrics (e.g logit difference) and KL Divergence (which was originally recommended). For the docstring task, we also include repeated activation patching as another point of reference for performance comparisons. We applied repeated activation patching by running the same circuit discovery method described in Section 3.3 but using Equation (1) rather than absolute attribution scores. Activation patching was not included in the other tasks as it was too computationally expensive to run on the GPT-2 small models used by IOI and Greater-Than. Subnetworks found using EAP for all three tasks are shown in Appendix A.

The ROC curves in Figure 2 suggest the performance of EAP is better than ACDC overall: it has the maximal AUC in Figure 2-2, while ACDC used

with the KL Divergence metric outperforms EAP in Figure 2.

## 4.2 Validating EAP Attribution Scores

In this section, we look at the approximate metric change (attribution score) EAP assigns to each edge in the model. We aim to understand the relation between the attribution score and the function of the edge in the task being studied. First, we look at the distribution of scores for edges in the circuit compared to edges not in the circuit for each of the three tasks.

Figure 3 shows the distribution of attribution scores for the IOI task. The distributions for the remaining tasks can be found in Appendix B. Qualitatively, attribution scores for edges in the circuit tend to be spread further from zero. Furthermore, there are only 6 edges outside of the interval $[-0.25, 0.25]$ that aren't part of the IOI circuit. We further explore the attribution scores for the IOI circuit's classes of heads in Appendix E.

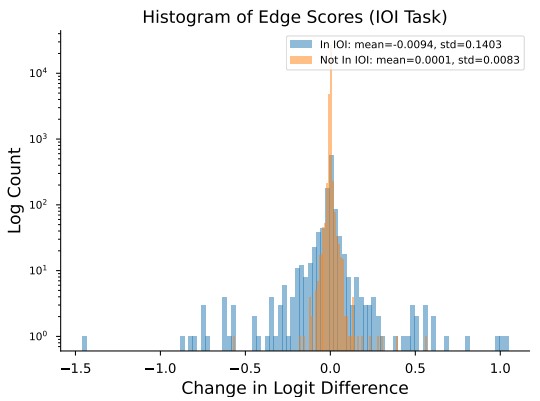

Figure 3: Distribution of Attribution Scores for the IOI Task (Logit difference metric)

## 5 Limitations

We introduced edge activation patching as an approximation to activation patching. However, we found that edge activation patching outperformed ACDC, a technique based on activation patching (Section 4). In this section, we investigate whether attribution patching's success is due to extremely accurate approximations (in Section 5.1 we find that the answer is no), and whether there is any further use for ACDC (in Section 5.2 we find that the answer is yes). We use the docstring task as a case study due to the small model size used.

## 5.1 How faithful are Attribution Patching's approximations?

To study how faithful the approximation Equation (2) is, we plot the attribution patching scores (Equation (2)) against the activation patching scores (Equation (1)) in Figure 4a. Surprisingly, we find a fairly weak correlation between activation and attribution patching scores ($R^2 = 0.27$). Further, the line of best fit has gradient 0.531, suggesting that attribution patching estimates the effect of activation patching as twice as important as it really is.

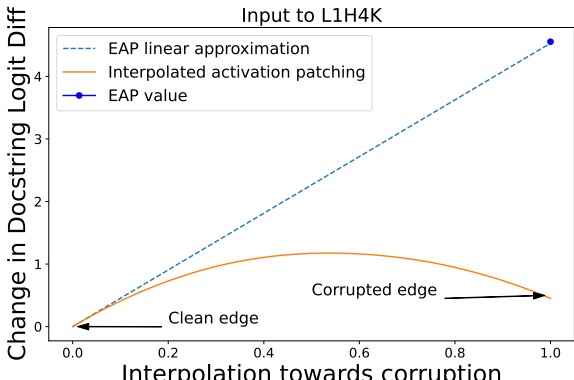

(a) Distribution of attribution scores for edges from activation patching and attribution patching. Circled: outlier EAP point studied in Figure 4b.

(b) Visualizing the rightmost point in Figure 4a. Note that corrupting this edge (surprisingly) slightly increases the logit difference on the Docstring task (higher logit difference is better). However, EAP overestimates how large this increase is.

Figure 4: Visualizing Edge Attribution Patching.

Moreover, we can gain some sense for the discrepancy between activation and attribution patching by studying the continuous transition between clean ($e_{\text{clean}}$) and corrupted ($e_{\text{corr}}$) activations in Equation (1), i.e studying the values $|L(x_{\text{clean}}| \text{ do}(E = \lambda e_{\text{corr}} + (1 - \lambda)e_{\text{clean}})) - L(x_{\text{clean}})|$ for $0 \leq \lambda \leq 1$. We can compare this to

the linear approximations of Attribution Patching $\lambda \Delta_e L$. Figure 4b shows the result for one edge in the docstring circuit where the linear approximation to activation patching is not accurate.

We find that interpolating towards the corrupted input creates a concave curve (Figure 4b) such that the linear approximation at $\lambda = 0$ overestimates the effect of activation patching this edge. In Appendix D we show that this also holds for the other outlier edges in the ellipse in Figure 4a.

### 5.2 Is there any further use for ACDC?

In Section 5.1 above, we found that EAP overestimates activation patching in cases where the attribution score is concave. This suggests the potential to refine the result by running ACDC on the pruned subgraph returned by EAP. We ran EAP first, then ACDC on the resulting subgraph for the Docstring task, varying pruning thresholds for EAP and ACDC independently. Figure 5 compares the TPR and FPR for the combined methods with the ROC curve of EAP only. The combined methods show increased performance compared to EAP only.

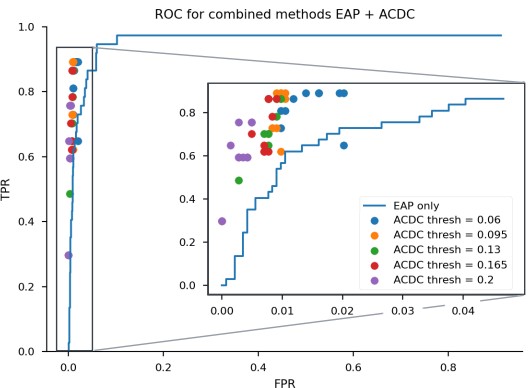

Figure 5: Comparing statistics of the combined EAP + ACDC methods with EAP only. The inset shows a zoom to the significant area of the statistics of the combined method.

Finally, one further limitation of this research is that the metrics used for interpretability do not precisely capture meaningful human understanding. Recovering a subgraph that humans previously recovered is limited because i) we can't evaluate this metric for interpretability tasks that we don't yet understand and ii) human-found circuits are imperfect, increasing the noise in this measurement.

## 6 Conclusion

We provide evidence that Edge Attribution Patching (EAP) outperforms ACDC in identifying circuits while being substantially faster to compute. This result is surprising, as EAP is an approximation for activation patching, the method applied by ACDC. However, running ACDC on the prepruned subnetwork found by EAP can improve the identification of relevant edges. Therefore, we suggest future circuit discovery experiments to run EAP first and then apply ACDC.

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

## A EAP Subnetworks

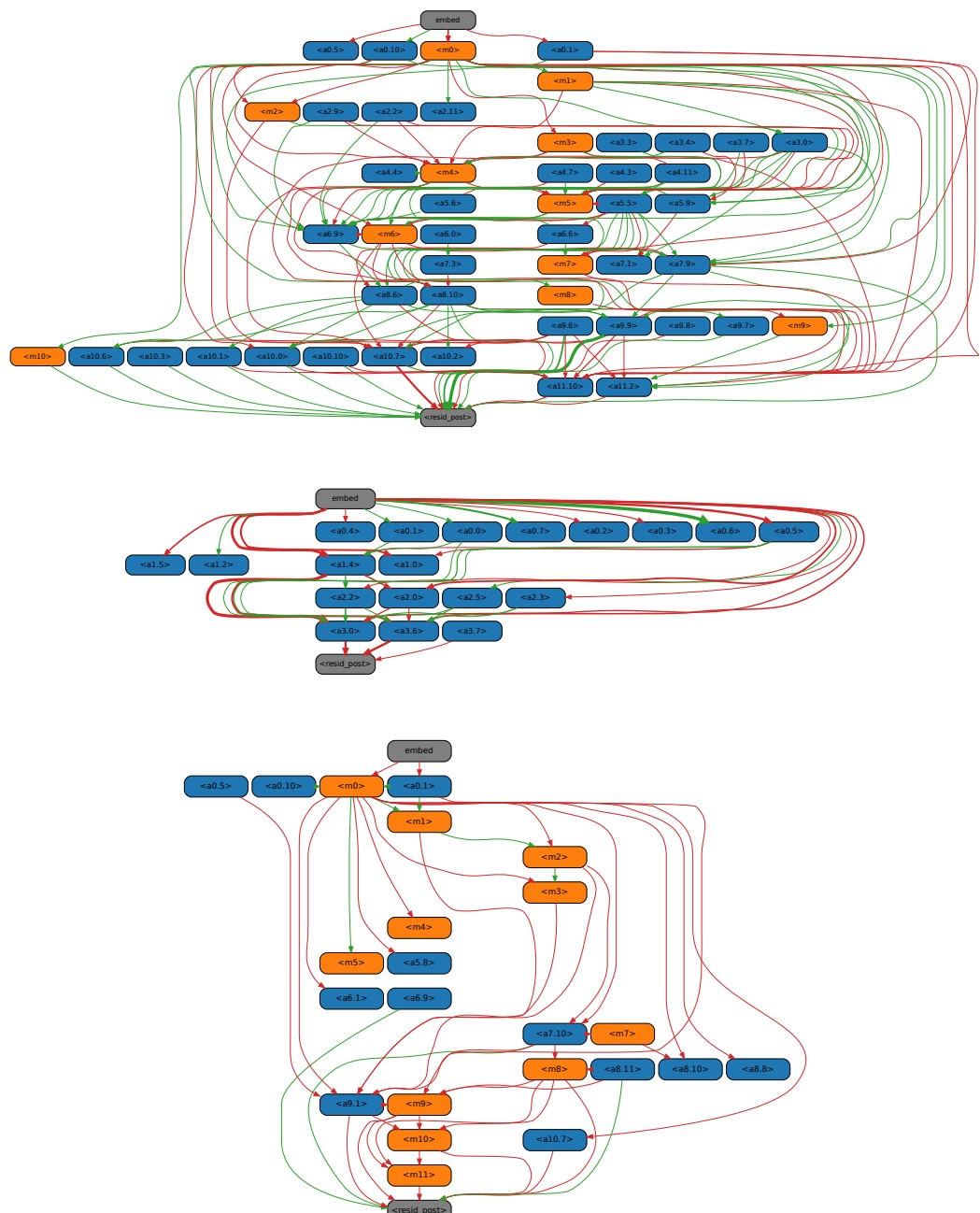

Figure 6: Resulting subnetworks after EAP at the given thresholds: (Top) IOI Subnetwork, Threshold=0.077; (Middle) Docstring Subnetwork, Threshold=0.244; (Bottom) Greater-Than Subnetwork, Threshold=0.009.

## B Distribution of EAP Attribution Scores

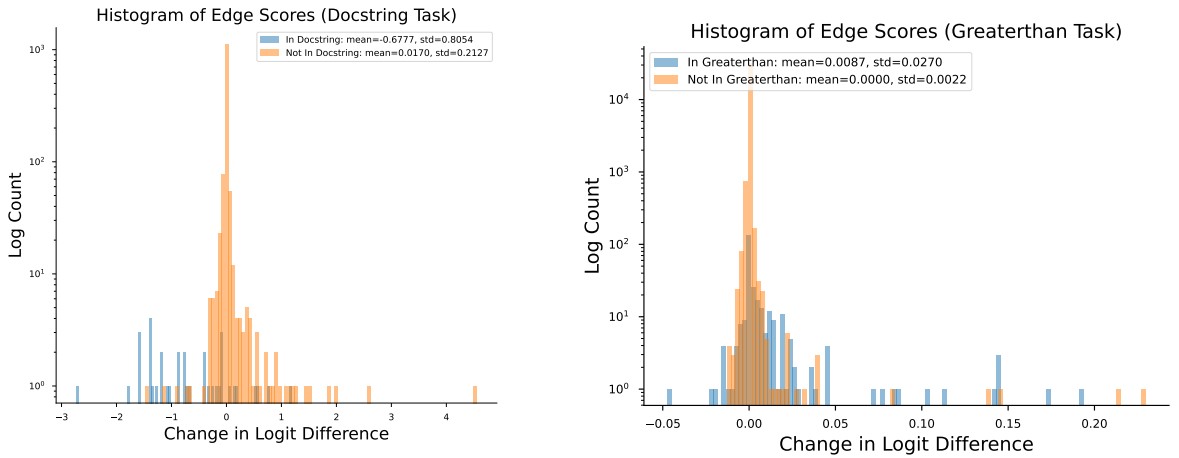

(a) Distribution of Attribution Scores for the Docstring Task    (b) Distribution of Attribution Scores for the Greater-Than Task

Figure 7: Distribution of Attribution Scores for the Docstring and Greater-Than tasks

## C Further investigation into combining EAP with ACDC

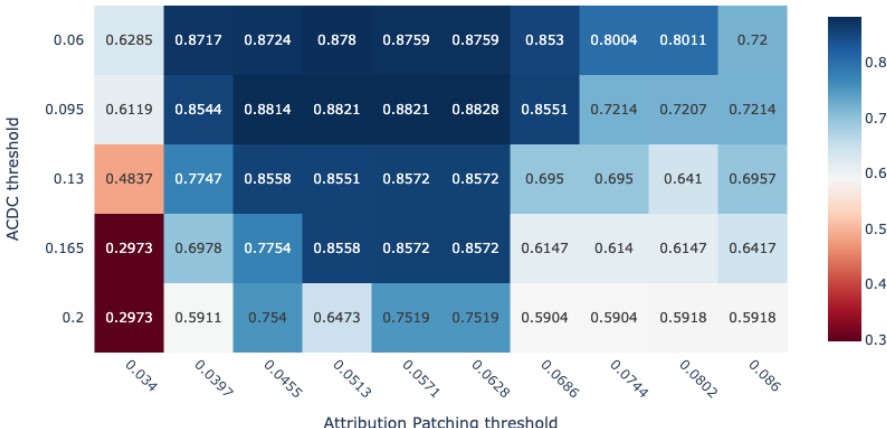

Figure 8: Youdens-J statistic (maximum TPR minus FPR value) for combining EAP and ACDC methods on the docstring task. We applied ACDC to the pruned subgraph returned by EAP.

## D Further failures of attribution patching approximation

In Figure 9 we show further cases where in the docstring task attribution patching can be misleading. These cases all involve an edge that comes from the model's embeddings (positional and tokens). Our interpretation is that weighted averages of embeddings are anomalous inputs to the model and cause the concave change in docstring logit diff which doesn't occur when edges ae between non-embedding model components.

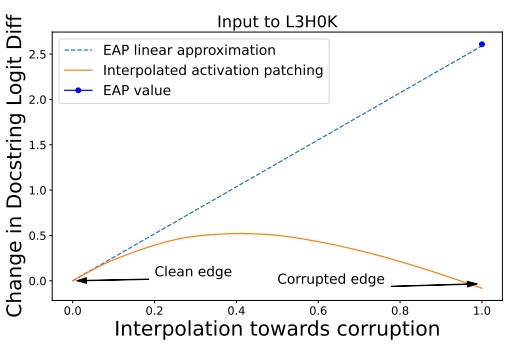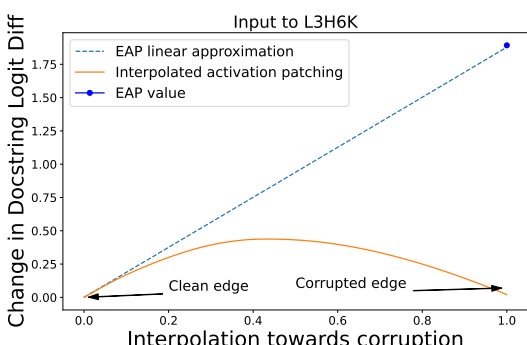

Figure 9: Visualizing Edge Attribution Patching in two further cases where the concave activation patching curve means the linear fit is poor.

# E  Edges Roles in IOI

We further explore the attribution scores for the IOI circuit. The IOI circuit is comprised of different attention head classes such as Induction heads, S-Inhibition heads, etc. (Wang et al., 2023). Figure 10 shows the distributions of scores stratified by the roles of the edges. The edge roles are defined according to the role of their origin node. While edge roles such as Previous Token, Duplicate Token, Induction, and S-Inhibition edges have attribution scores centered around zero, we see a bias in edge scores given to name mover and negative name mover edges. As the name mover edges are directly responsible for the model outputting the indirect object, the attribution scores are largely negative since ablating these edges removes the model's ability to output the indirect object, lowering the logit difference. Similarly, the negative name movers have attribution scores that are largely positive since ablating these edges improves the logit difference. This matches the intuitive function of the edges.

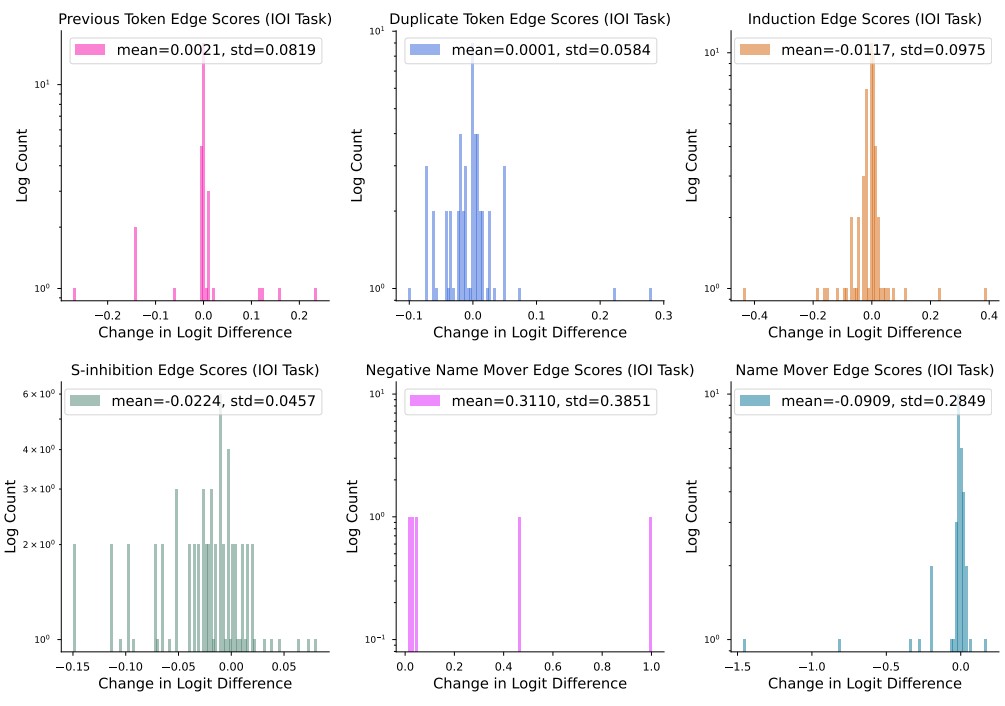

Figure 10: Distribution of Attribution Scores for each Edge Role in the IOI Task.

## F  Only one backwards pass is required for EAP

There are only two types of edges iterated over in ACDC: i) residual edges where the result is added at its endpoint, and ii) edges between the residual stream and the query, key and value calculations. Clearly for all edges like ii) we can compute the gradient terms in Equation (2) in one backwards pass.

Interestingly, for all $\Delta_e L$ terms where $e$ is a type i) edge (i.e added at the endpoint), we only need calculate the gradient with respect to the endpoint of the edge! For example, suppose we're calculating the effect of L0H0 on L1H0Q. If we represent the input to L1H0Q as a node $V$ in the computational graph then

$$\frac{\partial}{\partial e_{\text{clean}}} L(x_{\text{clean}}| \text{ do}(E = e_{\text{clean}})) = \frac{\partial}{\partial v_{\text{clean}}} L(x_{\text{clean}}| \text{ do}(V = v_{\text{clean}})) \tag{3}$$

due to how $V$ is just the sum of all the edges entering $V$. This allows efficient calculation of all the $\Delta_e L$ values since gradients with respect to nodes in computational graphs are calculated by default in backwards passes.