# OpenReview forum: "Attribution Patching Outperforms Automated Circuit Discovery"
_EMNLP/2024/Workshop/BlackBoxNLP — BlackboxNLP 2024_

### Official Review · Reviewer_HGi3 · 2024-08-24

**Overall Assessment:** 4
**Confidence:** 5

**Best Paper:**

2

**Best Paper Justification:**

I think this is a good paper which has influenced a lot of recent circuit-finding work; if it had been an archival submission one year ago, I might have nominated it for best paper. That said, the paper itself has some flaws, and is now a bit behind current work in the field.

**Comments Questions Suggestions And Typos:**

- Say a little more about circuits. How does mech interp view the computational graph? What are your nodes V (attn heads and MLPs) and edges E between them? What does this conceptualization of circuits capture (structure) and what does it miss (semantics)?
- Don't capitalize "Interchange Interventions", etc.; they're not proper nouns
- Turn Figures 1,2, 4a, and 5 into PDF/vector graphics figures with selectable text
- You refer to "Figure 2-2", but it's not clear which one that is

**Paper Summary:**

This paper extends an existing method for finding node-level circuits (attribution patching) to the edge domain, allowing circuits to be found at the edge level. It then conducts experiments that demonstrate that, despite taking significantly less time to run than another popular method (ACDC) the circuits it finds have equal or better overlap with known task circuits for the IOI, Greater-Than, and Docstring tasks. Surprisingly, though edge attribution patching is effective at finding circuits, it is not actually a good approximation of activation patching scores.

**Summary Of Strengths:**

This paper introduces a new method (EAP) for circuit-finding that is both much faster than the existing method (ACDC) and equally effective (or better). It also allows for the finding of circuits of various sizes using the scores EAP returns; ACDC would require a new run for each circuit size. While some aspects of evaluation are lacking, follow-up work has confirmed the usefulness of EAP, both alone and as a base to build new methods on.

**Summary Of Weaknesses:**

This paper has a number of weaknesses, namely that:
1. it does not evaluate against all the techniques that ACDC did
2. its evaluation is limited to a small number of known circuits, in small models
3. it doesn't measure faithfulness
4. it is lacking a bit of background on what circuits are, to make this paper more accessible to readers new to circuits

However, this paper is also almost a year old, and has been succeeded by a number of other papers / circuit-finding methods (combining this method with integrated gradients, or using differentiable masking to find circuits). These papers have addressed much of 2-3 (and 1 looks in retrospect to be less important). So, it's not clear to me that these flaws are very important now. However, I would love to see the background on circuits be explained more clearly, and to see the comments below be addressed.

---

### Official Review · Reviewer_Rf8b · 2024-09-08

**Overall Assessment:** 4
**Confidence:** 4

**Best Paper:**

1

**Best Paper Justification:**

NA

**Comments Questions Suggestions And Typos:**

NA

**Paper Summary:**

This paper examines the automated interpretability of large NNs. It compares the SOTA Automatic Circuit discovery(ACDC) algorithm with a new technique called attribution patching (Nanda, 2023). The system uses the scores from the attribution patching algorithm to keep the top K edges of an NN. This work conducts empirical experiments to compare the performance of this new technique with more rigorous but computationally expensive ACDC method.

**Summary Of Strengths:**

1. This paper provides interesting empirical results on attribution vs activation patching approaches.
2. It finds that attribution patching is more accurate than ACDC in identifying circuits on 3 tasks overall. However, ACDC does much better in finding relevant edges within those circuits.
3. The paper suggests using attribution patching first to cut larger NNs into smaller circuits and then using ACDC within those circuits to find relevant edges more accurately.

**Summary Of Weaknesses:**

1. This paper does not introduce any new methodology or system for NN interpretability. It seems to apply a sorting algorithm to the existing attribution patching algorithm.
2. The abstract and introduction should be rewritten to state this paper’s contributions clearly. An initial read may suggest that attribution patching is a contribution of this work, but that was proposed in Nanda, 2023.

---

### Official Review · Reviewer_KRi9 · 2024-09-10

**Overall Assessment:** 4
**Confidence:** 4

**Best Paper:**

1

**Best Paper Justification:**

not applicable

**Comments Questions Suggestions And Typos:**

This work uses less than the maximum allowed number of pages, indeed it has only 5 pages. So I recommend the authors use the full paper length of 8 pages to clarify a number of points in their camera-ready version, if accepted for publication.

- In the abstract you write that your approach requires only 2 forward passes and 1 backward pass. While I understand that 1 backward pass is required to compute gradients, it is less clear to me why 2 forward passes are needed. Could you please clarify what is done in each of theses forward passes?

- page 2 Related Work on Automated Circuit Discovery: "that model used" -> "that THE model uses", "slow forward passes" -> "USE slow forward passes" (Btw why are these forward passes slow? Is there some overhead due to storing some intermediate activations during the forward pass?)

- Please specify the tasks and models used in your experiments (no reference is given for them so far)

- Page 3 end of section 4.1: Please clarify when EAP is better than ACDC, and when the opposite is true. Does this depend on the metric used to find important edges? Why is it so?

- Section 5 first sentence: "We introduced edge activation patching" -> "We introduced edge ATTRIBUTION patching"

- Section 5.2 first sentence: "... in cases where the attribution score is concave" -> "... in cases where the CHANGE IN LOGIT DIFFERENCE METRIC is concave"

- What is the threshold you apply to obtain the ROC curves? Is it the value of the edge importance above which edges get selected for the circuit?

- In figure 4b and 9 you show examples where the linear approximation to activation patching is not accurate. Could you please also show examples where the approximation IS accurate?

- In my understanding section 4.2 is kind of superfluous since you show there that the edges selected to be in the circuit have higher absolute attribution scores. But this is exactly how you choose the edges to keep in the circuit in section 3.3. So this result seems not surprising. Maybe remove that section or justify why it is important and not redundant with your edge selection approach.

**Paper Summary:**

This work addresses the problem of automated circuit discovery, i.e. of identifying a subnetwork (circuit) of a model that is responsible for solving a specific subtask.

To this end the authors propose to prune unimportant edges in the computational graph of the neural network based on gradient-based estimation of edge importance. They call this new approach Attribution Patching (since they use a standard gradient-based XAI attribution method to estimate the edges' importance).

The authors apply their method to transformer-based language models on three tasks (completion of docstrings, Indirect Object Identification IOI, greater-than task) and show it performs better in terms of the AUC for identifying edges in the circuit than Automated Circuit Discovery (ACDC) from Conmy et al. 2023 which is based on Activation Patching.

The advantage of the proposed method is that it is less expensive to compute than previous methods based on iteratively editing the network's connections to assess each edge's importance in order to prune the network to obtain the circuit. The authors further show that this new approach can also be complementary to ACDC, indeed they show that when Attribution Patching is applied as a first step before applying ACDC, then this gives the best results.

**Summary Of Strengths:**

- A new way of finding circuits in neural networks in an automated way based on an XAI attribution technique, namely gradient-based attribution. So it combines mechanistic interpretability with attribution-based XAI.

- Evaluation on 3 different tasks, demonstrating the usefulness of the method.

- The approach is more efficient than ACDC from Conmy et al. 2023, it requires only two forward passes and one backward pass, as opposed to multiple forward passes in ACDC.

**Summary Of Weaknesses:**

- The work is almost identical to an approach proposed in an "incomplete project report" posted by Neel Nanda in February 2023: https://www.neelnanda.io/mechanistic-interpretability/attribution-patching
Although this report (blog post) is not an official peer-reviewed publication, and the authors do cite it, it shall be more clearly acknowledged by the authors due to the great similarity to the present work IMHO.

- Another recent work by Ferrando et al. goes in a similar fashion of finding circuits in an automated way using an XAI method, namely ALTI (Aggregation of Layer-Wise Token-to-Token Interactions) XAI method: https://arxiv.org/pdf/2403.00824
You might want to cite this work too.

---

### Decision · Program_Chairs · 2024-09-20

**Decision:**

Accept

**Comment:**

All the reviewers agree that this paper introduces a very valuable interpretability method that will have a positive impact on the field. As such we gladly accept it to BlackboxNLP.

The reviewers, however, do raise several salient points for improvement that the authors are strongly encouraged to engage with in the camera ready version of the paper. Reviewer KRi9 provides various suggestions for improving the writing style and presentation of the paper that should be incorporated. Furthermore, given that the paper is currently only 5 pages, the authors are encouraged to take up the suggestion of reviewer HGi3 of incorporating a more detailed background on circuit finding methods.